# Malaria and intestinal parasite co-infection and its association with anaemia among people living with HIV in Buea, Southwest Cameroon: A community-based retrospective cohort study

Sorelle Mekachie Sandie[1]*, Irene Ule Ngole Sumbele[1,2], Martin Mih Tasah[1], Helen Kuokuo Kimbi[1,3]

1 Department of Zoology and Animal Physiology, University of Buea, Buea, Cameroon, 2 Department of Microbiology and Immunology, College of Veterinary Medicine, Cornell University, Ithaca, New York, United States of America, 3 Department of Medical Laboratory Science, The University of Bamenda, Bambili, Cameroon

☯ These authors contributed equally to this work.
* sandiesorelle2@gmail.com

**Data Availability Statement:** All relevant data are within the manuscript.

## Abstract

### Background

Both malaria and intestinal parasites are endemic in Cameroon, and their co-infection can be of great impact on anaemia among people living with HIV (PLWH). This community-based retrospective cohort study determined the prevalence and association of infections with anaemia in PLWH and HIV-negative individuals in Buea, Cameroon from March to August 2019.

### Methods

The study population comprised of 190 PLWH and 216 consenting HIV-negative individuals from the Buea community. Participants were examined clinically, the collected blood sample was used for malaria parasite (MP) detection, HIV diagnosis and haemoglobin (Hb) measurement while stool samples were examined for the detection of intestinal parasites (IPs). Proportions were compared using Pearson's Chi-square test and association of anaemia with independent variables was evaluated using logistic regression analysis.

### Results

Out of the 406 participants, MP, IPs and MP/IP co-infection prevalences were 15.5%, 13.0% and 3.0% respectively. PLWH had a higher prevalence of MP (16.3%, P = 0.17), IPs (23.7%, P < 0.001) and MP/IPs co-infection (3.7%, P = 0.04) when compared with HIV-negative participants. Similarly, PLWH had significantly lower mean haemoglobin value (11.10 ± 1.54 g/dL) than their HIV-negative counterparts (12.45 ± 2.06 g/dL). Also, PLWH co-infected with MP and IPs were observed to have a significantly lower mean haemoglobin value (10.6 ± 1.21 g/dL). PLWH had a significantly (P < 0.001) higher prevalence of mild

**Funding:** The author(s) received no specific funding for this work.

**Competing interests:** The authors have declared that no competing interests exist.

(56.8%), moderate (18.4%) and severe (1.6%) anaemia when compared with HIV-negative counterparts. The significant risk factors associated with anaemia included being febrile (P = 0.03), MP-infected only (P = 0.001), HIV-infected only (P < 0.001), having dual (P < 0.001) or triple-infections (P = 0.03).

## Conclusion

Malaria and intestinal parasites remain public health concerns among PLWH and anaemia as a serious haematological abnormality gets exacerbated even with the viral load suppression. Hence, routine medical check-ups among PLWH are recommended.

## Background

Parasitic infections are a major cause of morbidity and mortality in Africa; especially in resource-limited tropical and sub-tropical regions like sub-Saharan Africa [1] where most HIV/AIDS cases are concentrated. As recently reported by UNAIDS in 2019, there were roughly 1.7 million new HIV infections, which is an insignificant improvement when compared to 2017 (1.8 million) [2]. In the same year, Cameroon reported an HIV incidence of 1.02 and a national prevalence of 3.6% [2]. People living with HIV (PLWH) are threatened by a great number of diseases caused by different types of biological agents. Although the morbidity and mortality due to malaria and intestinal parasites can be controlled by the application of various prevention measures put in place by the Cameroon government as well as the delivery of chemotherapy, elimination and final eradication will not be achieved until populations live in mosquito-free environments and have access to effective sanitation, sewage treatment and waste disposal; which remain a trending problem in Cameroon.

HIV infection is characterized by immunosuppression that exposes the infected individual to a variety of microbes such as *Plasmodium*, *Mycobacterium* and several intestinal parasites [3]. These microbes (opportunistic or not) are a serious threat to the health status of people living with HIV, especially in developing countries like Cameroon. In an attempt to achieve HIV epidemic control in 2021, the Cameroon Government has been implementing various strategies that led to an encouraging increase in antiretroviral therapy (ART) coverage (53%) and viral load (VL) suppression (44.7%) nationally in 2019 [4].

African countries still bear the largest burden of malaria morbidity (93%), with *Plasmodium falciparum* accounting for 99.7% of malaria cases in 2019 [5]. In 2018, WHO reported that Cameroon featured the fourth country behind Nigeria, Democratic Republic of Congo and Mozambique with the highest malaria prevalence [6]. Many studies have consistently reported an association between HIV infection and malaria in Cameroon [7–11], with each region showing a varied prevalence of both malaria and HIV infection. Malaria transmission is stable throughout the year, with an overall malaria test positivity rate of 62% registered in 2017 [12].

The burden of intestinal parasite infections (IPIs) is greatest in low-income areas and are major public health problems in certain areas of Sub-Saharan Africa. These IPIs are associated with a humid climate, unsanitary environments and poor socio-economic conditions. Parasites causing these IPIs are protozoans (*Cryptosporidium species*, *Cyclospora cayetanensis*, *Isospora belli*, *Microsporidia spp* and *Entamoeba histolytica*) and helminths (*Ascaris lumbricoides*, *Trichuris trichiura*, hookworms, *Schistosoma mansoni* and *Taenia species*). *Cryptosporidium species*, *Cyclospora cayetanensis*, *Isospora belli*, and *Microsporidia spp* are often referred to as opportunistic parasites in people living with HIV because they cause great morbidity especially

when CD4 T cell count falls below 200 cells/μL [13]. Various studies carried out among African adults living with HIV have reported IPI rates ranging from 10% to 45% [14, 15]. Majority of the research on the impact of IPIs on HIV progression in Cameroon, occurred before the wide availability of ART [16] hence, the need for further evaluation.

In Cameroon, PLWH face similar environmental realities as those not infected with HIV and so can easily get co-infected with combinations of intestinal and malaria parasites. Such co-infections may have considerable health consequences which could lead to severe clinical symptoms and pathology in PLWH. Studies have shown that individuals with multiple infections have an increased risk of developing more frequent and severe diseases due to interactions among infecting parasite species [17–19].

Malaria and intestinal parasites in PLWH can both lead to severe haematological abnormalities. Although ART in combination with cotrimoxazole confers a level of immunity to infections in PLWH, it has been reported to have adverse effects on the red blood cell indices of these patients, thereby causing anaemia [20]. Although malaria and intestinal parasites, as well as ART, are known etiological factors in anaemia [11, 16, 21, 22], the extent to which their interaction might enhance the risk of anaemia merits further investigations. Previous studies have reported on the influence of malaria/HIV co-infection on the occurrence of anaemia [11, 19, 23]. Co-infections in HIV are numerous and important causes of morbidity and mortality. Combating co-infections has been identified as an important public health goal. The situation regarding PLWH co-infected with malaria and intestinal parasites in the area following the intensification of different control strategies such as free availability of ART and cotrimoxazole, access to viral load testing and community ART dispensation is unknown. Therefore, the present study was undertaken to determine the prevalence of malaria parasite and intestinal parasites in PLWH compared to HIV negative counterparts as well as to assess the effects of these infections on anaemia to provide relevant information to policymakers in the country.

## Methods

### Study area and participants

The study was carried out in the HIV care and treatment centre at the Regional Hospital and different neighbourhoods in the Buea community. The study area has been described in detail by Sumbele et al. [24]. The Buea Regional Hospital is the referral HIV treatment centre in Fako Division serving the population and hence receives PLWH daily from the various neighbourhoods for their monthly distribution of ART. With the presence of the Mount Cameroon, Buea is a major touristic site and has a business-friendly environment. In the latest published national HIV prevalence ranking, the South West Region is the 7th region with a prevalence of 3.6% which equates the national prevalence (3.7%) [25] as well as the second to the last Region (33.8%) with the least viral load suppression. Buea is an area of moderate malaria transmission, with a peak time for malaria transmission during and just after the rainy season (March to September) [26]. Its tropical climate provides a suitable environment for the development of the malaria vectors and this, therefore, contributes to increased malaria transmission. In addition, the inadequate access to safe and potable water in the region and Cameroon at large predisposes the population to water-borne infections including IPIs [27].

The two categories of participants, PLWH and HIV-negative individuals aged 1–72 years old were enrolled in the study and the HIV status of the HIV negative participants was determined using the Determine HIV Test strip. HIV-negative participants included persons who consented to be screened for HIV during the HIV outreach programmes and were found negative while PLWH were enrolled from the HIV care and treatment centre of the Buea Regional Hospital as they came for their ART monthly appointment. PLWH identified through the

outreach programmes in the community were referred to the HIV care and treatment centre for confirmation of their HIV positive status and follow up before being enrolled among the PLWH category. Only participants who signed the informed consent form and willingly accepted to be counselled and tested for malaria, IPIs and HIV were enrolled in the study. Participants on antibiotics, malaria treatment or antiparasitic agents two weeks before enrolment were excluded from the study.

## Study design and sample size

This study was a community-based retrospective cohort study carried out between April 2018 and August 2019. A purposive convenient method of sampling was employed and participants were enrolled through HIV outreach programmes in their homes and offices in the Buea neighbourhoods (Molyko, Bonduma, Great Soppo, Government reserve area (GRA), Buea town, Bokwaongo, Bokova, Mile 18, Clerks and Federal quarters). The sample size was determined using the formula $N = 2(Z_\alpha + Z_\beta)^2 P(1-P)/(P_0-P_1)^2$ [28] where N represented the sample size evaluated, $Z_\alpha$ was 1.96 and $Z_\beta$ was 0.84 which are the standard normal deviates (for the 95% confidence interval, CI), $P_0$ (Previous prevalence of MP/IPs co-infection in PLWH) was 50% and $P_1$ (Previous prevalence of MP/IPs co-infection in HIV negative individuals) was 67.2% [29]. The sample size calculated was 128 for each group giving an optimum sample size of 256 participants for the study.

## Data collection

**Questionnaire.** A pre-structured questionnaire was used to collect socio-demographic information such as age, sex, location of residence, level of education, marital status and occupation while clinical data such as ART usage, duration on treatment, ART regimen and viral load (VL) were collected from each HIV patient's file at the HIV care and treatment centre. Health and preventive practices towards malaria and IPIs as well as behavioural habits regarding health conditions with a history of symptoms were recorded.

**Sample collection and processing.** Participants were instructed on how to collect stool sample in universal transparent faecal containers with a screw cap lid. After the provision of a stool sample, the participant's body temperature was measured using a digital thermometer and fever was defined as temperature $\geq 37.5°C$. Blood was collected under sterile conditions into a well-labelled Ethylenediaminetetraacetate (EDTA) tube for each participant for HIV screening and confirmation, malaria diagnosis and haemoglobin (Hb) measurement. The collected samples were transported to the Malaria Research Laboratory of the University of Buea for further analysis.

On arrival in the laboratory, the stool samples were analysed macroscopically and preserved in 10% formalin. Formol-ether concentration technique and Modified Ziehl-Neelsen (ZN) methods [30] were performed for the detection of intestinal parasites (protozoans and helminths). To estimate parasite intensity, egg counts per slide were converted to egg per gram of faeces (epg) by multiplying the number of eggs on the slide by 24. The blood sample was used to prepare thick and thin blood films on a labelled slide that was Giemsa-stained, analysed following standard procedures and the parasite density per microliter of blood was calculated based on the number of parasites per 200 leucocytes on the thick blood film, assuming a leucocyte count of 8000 leucocytes/μL of blood as described by Cheesbrough [30]. All blood and stool slides were examined by two microscopists who were blinded to the results, and where positive/negative discrepancy occurred, slides were counter-read by a third microscopist, who was blinded regarding the two previous results. A negative result was declared after checking at least 200 microscope fields. The blood was also used for HIV rapid test (in the case of

presumed HIV-negative participants) and Hb measurement. Viral loads were obtained from the HIV positive participant's file at the HIV Care Centre. Viral load < 50 copies /mL of blood was considered undetectable [31].

CD4T cell counts were determined using the BD FACS Count™ System (BD Biosciences, USA) as per the manufacturer's instructions. To assure the quality of the laboratory data, standard operating procedure for each test was followed. Reagents were checked for expiry date and prepared according to the manufacturer's instruction. The Determine Rapid Test Kit (Abott Laboratories, CO., Ltd. Minato-Ku, Tokyo Japan) was used to test the supposed HIV negative subjects for HIV-1 following the manufacturer's instructions.

**Haemoglobin (Hb) measurement.** The full blood count (FBC) analysis was carried out using the URIT-3300 automated haematology analyser (URIT Medical Electronic CO., LTD. Jiuhua Road, Gungxi, China), following the manufacturer's instructions. Haemoglobin concentration was obtained from the FBC results. Anaemia was defined as Hb concentrations below the World Health Organization (WHO) reference values corresponding to age or gender [32]. Further classification was done to determine severe (Hb < 7g/dL), moderate (Hb between 7.0 and 10.0 g/dL) and mild anaemia (> 10 g/dL Hb < 11g/dL) cases [29].

**Statistical analysis.** Data were analysed using IBM-Statistical Package for the Social Science (SPSS) version 20.0 software (SPSS Inc., Illinois, USA). Baseline characteristics were summarized as mean with 95% CI and compared using the independent *t*-test. Malaria and intestinal parasites data were summarized into means and standard deviation (SD) and percentages were used in the evaluation of the descriptive statistics. The significance of differences in prevalence was explored using Pearson's Chi-square while Analysis of Variance (ANOVA), Kruskal-Wallis H test and Student's t-test was used where appropriate to assess difference in group means. The association of anaemia with the independent variables was evaluated using logistic regression model. $P < 0.05$ was considered for statistical significance.

**Ethical considerations.** Using an information sheet, a brief talk was given to the participants in the English Language (and exceptionally in Pidgin English) and the participants were then invited to participate in the study by signing an informed consent form. In the case of children ($\leq$ 15years), their parents signed proxy informed consent form on their behalf. Ethical clearance was obtained from the Institutional Review Board hosted by the Faculty of Health Sciences of the University of Buea (Reference Number: 661–09). For confidentiality purposes, patients were identified by special identification numbers. Results of participants with parasitic infections in the neighbourhoods were referred to a clinician for their case management.

## Results

### Socio-demographic and clinical characteristics of the participants

Five hundred and eighty-nine (589) participants were approached to participate in the study, those who could not provide stool sample and whose blood sample could not be collected were excluded from the study; giving a total of 406 (68.9%) successfully enrolled participants. Out of the 406 participants, 190 were PLWH and 216 were HIV-negative from the communities as shown in Fig 1.

Overall, the mean ± SD age of the participants was 31.2 ± 16.7 (range = 1–70) years. There were more females (68.5%) than males (31.5%). A greater proportion of the study population belonged to the > 40 years age group (31.0%) and the least to the 31–40 years age group (18.7%). Majority of the PLWH had primary level of education (66.7%), were self-employed (31.1%) and were single (62.0%). Among HIV negative participants, majority had university level of education (35.2%), were students (56.5%) and single (63.0%) as revealed in Table 1.

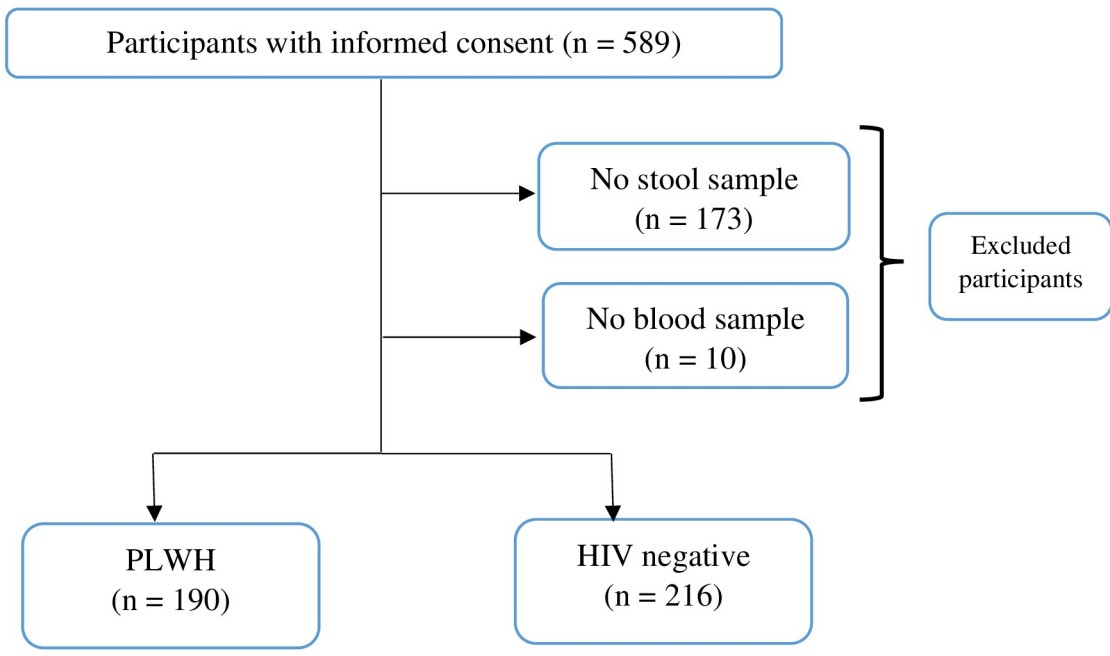

**Fig 1. Study flow diagram showing compliance of the participants.**

Table 1. Characteristics of the study population.

| Parameter | Category | PLWH | HIV Negative | Overall | $\chi^2$ |
|---|---|---|---|---|---|
| | | n (%) | n (%) | n (%) | P-value |
| Age group (years) | ≤ 20 | 42 (22.1) | 72 (33.3) | 114 (28.1) | |
| | 21–30 | 20 (10.5) | 70 (32.4) | 90 (22.2) | 48.64 |
| | 31–40 | 48 (25.3) | 28 (13.0) | 76 (18.7) | < 0.001* |
| | > 40 | 80 (42.1) | 46 (21.3) | 126 (31.0) | |
| Mean age ± SD (years) | | 35.5±16.5 | 27.6±15.8 | 32.3±16.7 | |
| Sex | Female | 140 (73.7) | 138 (63.9) | 278 (68.5) | 4.49 |
| | Male | 50 (26.3) | 78 (36.1) | 128 (31.5) | 0.02* |
| Educational Level | No formal | 6 (3.7) | 18 (8.6) | 24 (6.0) | |
| | Primary | 126 (66.3) | 54 (25.7) | 180 (45.1) | 78.94 |
| | Secondary | 43 (22.6) | 64 (30.5) | 107 (26.8) | < 0.001* |
| | Tertiary | 14 (7.4) | 74 (35.2) | 88 (22.1) | |
| Occupation | Student | 48 (25.4) | 122 (56.5) | 171 (42.1) | |
| | Unemployed | 25 (13.2) | 12 (5.2) | 37 (9.1) | 45.38 |
| | Self-employed | 59 (31.2) | 28 (13.0) | 87 (21.4) | < 0.001* |
| | Employee | 57 (30.2) | 54 (25.0) | 111 (27.3) | |
| Marital status | Single | 93 (62.0) | 102 (63.0) | 197(62.5 | 0.03 |
| | Married | 61 (38.0) | 60 (37.0) | 117(37.5) | 0.47 |
| ITN usage | Yes | 109 (57.4) | 76 (35.2) | 185 (45.6) | 20.05 |
| | No | 81 (42.6) | 140 (64.8) | 221 (54.4) | < 0.001* |

**Table 2. Clinical characteristics and laboratory profile of the study population.**

| Parameters | | PLWH n (%) | HIV Negative n (%) | Overall n (%) | $\chi^2$ P-value |
|---|---|---|---|---|---|
| Fever status | Febrile | 23 (12.1) | 34 (15.7) | 57 (14.0) | 1.12 |
| | Afebrile | 167 (87.9) | 182 (84.3) | 349 (86.0) | 0.18 |
| Mean temperature ±SD (˚C) | | 37.1 ±0.5 | 37.1 ±0.5 | 37.1 ±0.5 | |
| Malaria clinical signs | Yes | 39 (23.5) | 32 (18.2) | 71 (20.7) | 1.40 |
| | No | 127 (76.5) | 144 (81.8) | 272 (79.3) | 0.15 |
| Anaemic status | Anaemic | 146 (76.8) | 86 (39.8) | 232 (57.1) | 56.59 |
| | Non-anaemic | 44 (23.2) | 130 (60.2) | 174(42.9) | < 0.001* |
| Mean Hb ±SD (g/dL) | | 11.1±1.5 | 12.5 ±2.1 | 11.8 ±1.9 | |
| CD4 T-cell count(cells/μL) | < 200 | 15 (14.2) | - | 15 (14.2) | - |
| | 200–500 | 51 (48.1) | - | 51 (48.1) | |
| | >500 | 40 (37.7) | - | 40 (37.7) | |
| Mean CD4 T cell count ±SD (cells/μL) | | 466.0 ±275.9 | - | 466.0±275.9 | |
| ART usage | Yes | 186 (97.9) | - | 186 (97.9) | |
| | No | 4 (2.1) | - | 4 (2.1) | |
| Zidovudine-based ART | Yes | 171 (91.9) | - | 171 (91.9) | |
| | No | 15 (8.1) | - | 15 (8.1) | |
| ART Duration (years) | 0 | 4 (2.1) | | 4 (2.1) | |
| | < 1 | 36 (19.4) | - | 36 (19.4) | |
| | 1–3 | 61 (32.8) | - | 61 (32.8) | |
| | >3 | 89 (47.8) | - | 89 (47.8) | |
| Viral load (RNA copies/ml) | Not detectable (<50) | 141 (77.5) | - | 141 (77.5) | |
| | ≥ 50 | 41 (22.5) | - | 41 (22.5) | |

Less than half of the study population (45.6%) used ITNs as a preventive measure against malaria. A greater proportion of PLWH used ITNs (57.4%) more than the HIV negative counterparts (35.2%). As seen in Table 1, there were statistically significant differences in the various variables between PLWH and HIV- participants apart from marital status.

Axillary temperature measurement revealed that 14.0% (57) of the general population were feverish, with the highest proportion (15.7%) among HIV-negative participants. Malaria clinical signs (headache, joint pains, nausea, vomiting etc...) were less frequent among HIV-negative participants than among PLWH (23.4%). Anaemia occurred in 57.1% (232) of the participants and PLWH had the highest occurrence (76.8%) when compared with their HIV negative counterparts and the difference was statistically significant ($\chi^2$ = 56.59, P < 0.001). Among PLWH, half of the population (50.0%) had CD4 T cell count between 200–500 cells/μL of blood, the majority (97.9%) were on ART, 91.9% (171) were taking zidovudine-based ART and 77.5% (141) had attained viral suppression(< 50 copies/mL of blood) as shown in Table 2.

## Malaria parasite, intestinal parasite infection and MP/IPI co-infection prevalence

Out of the 406 participants, 63 (15.5%) were positive for MP with *Plasmodium falciparum* as the only species observed. MP prevalence was higher in PLWH (16.3%) than those negative (14.8%) although the difference was not statistically significant (P = 0.39). Overall, 14.0% (57) of the participants were positive for IPIs with a higher occurrence of protozoans (10.4%) than helminths (3.8%). The most prevalent protozoan parasite was *Cryptosporidium* species while that of helminths was hookworm. The prevalence of IPs was significantly

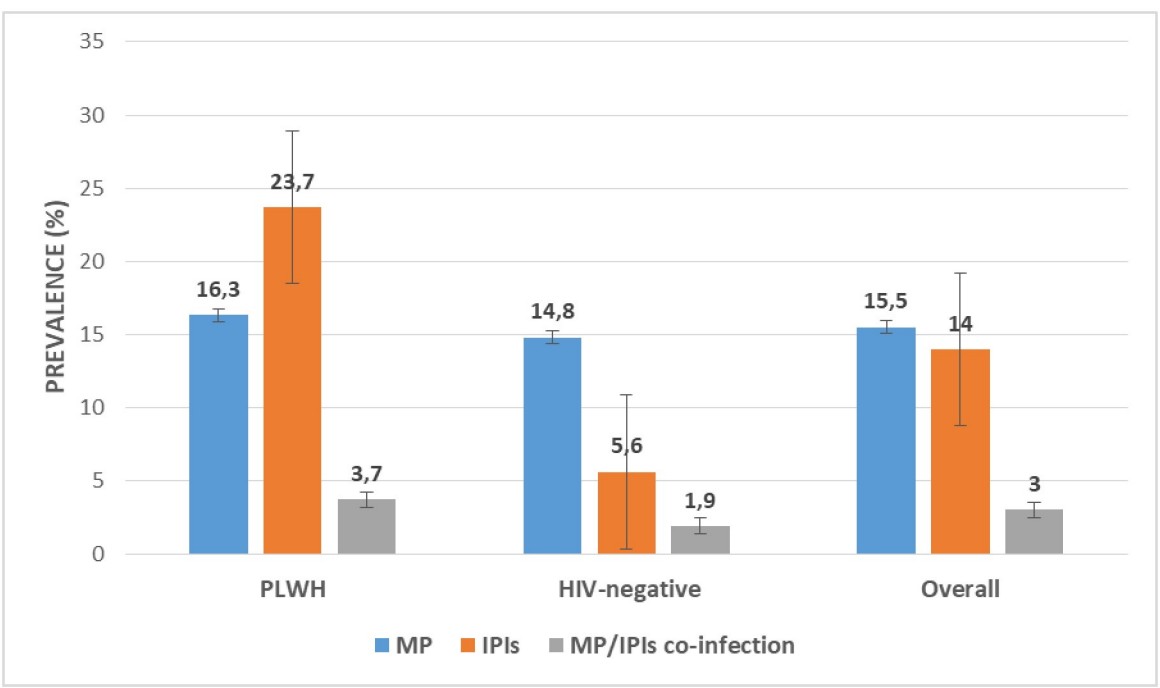

**Fig 2. Prevalence of MP, IPIs and MP/IPIs co-infection in the study population.**

higher ($\chi^2$ = 31.73, P < 0.001) among PLWH (23.7%, 45) than in those negative (5.6%, 12). Malaria/IP co-infection prevalence was 2.7% (11) and although the difference was not significant (P = 0.20), malaria/IP co-infection was higher in PLWH (3.7%) compared to HIV-negative participants (1.9%) as shown in Fig 2.

## Infection categories, haemoglobin level and anaemia prevalence

Out of the 406 participants, 1.7% (7) had triple infection (MP, HIV and IPIs), 9.4% (38), 5.9% (24) and 1.0% (4) were co-infected with IPIs/HIV, MP/HIV and MP/IPIs respectively. Equally, 29.8% (121) had HIV only, 6.9% (28) had malaria only and 2.0% (8) had IPIs only. The rest of the participants (43.3%, 176) were uninfected. As shown in Fig 3, there was a significant difference (H = 68.24, P < 0.001) when comparing mean haemoglobin values between the different infection categories; as those triply infected had the lowest mean haemoglobin value (10.55 ± 1.20g/dL) when compared with dual and singly infected patients accordingly.

The prevalence of anaemia in the study population was 57.1% (232) and PLWH had a significantly higher ($\chi^2$ = 56.59, P < 0.001) prevalence of anaemia (76.8%) than their HIV-negative counterparts (39.8%). When comparing single infections, the prevalence of anaemia was highest among those who had HIV only (79.3%), followed by those infected with malaria parasite only (71.4%) and then those with IPs only (25.0%) and the difference was statistically significant (P = 0.002). Participants infected with MP, HIV and IPs simultaneously were observed to have the highest prevalence of anaemia (85.7%) when compared with other co-infection categories as shown in Table 3.

With respect to anaemia severity, PLWH had a significantly (P < 0.001) higher prevalence of mild (56.8%), moderate (18.4%) and severe (1.6%) anaemia when compared with HIV-negatives. Severity of anaemia with respect to infection categories is shown in Table 3.

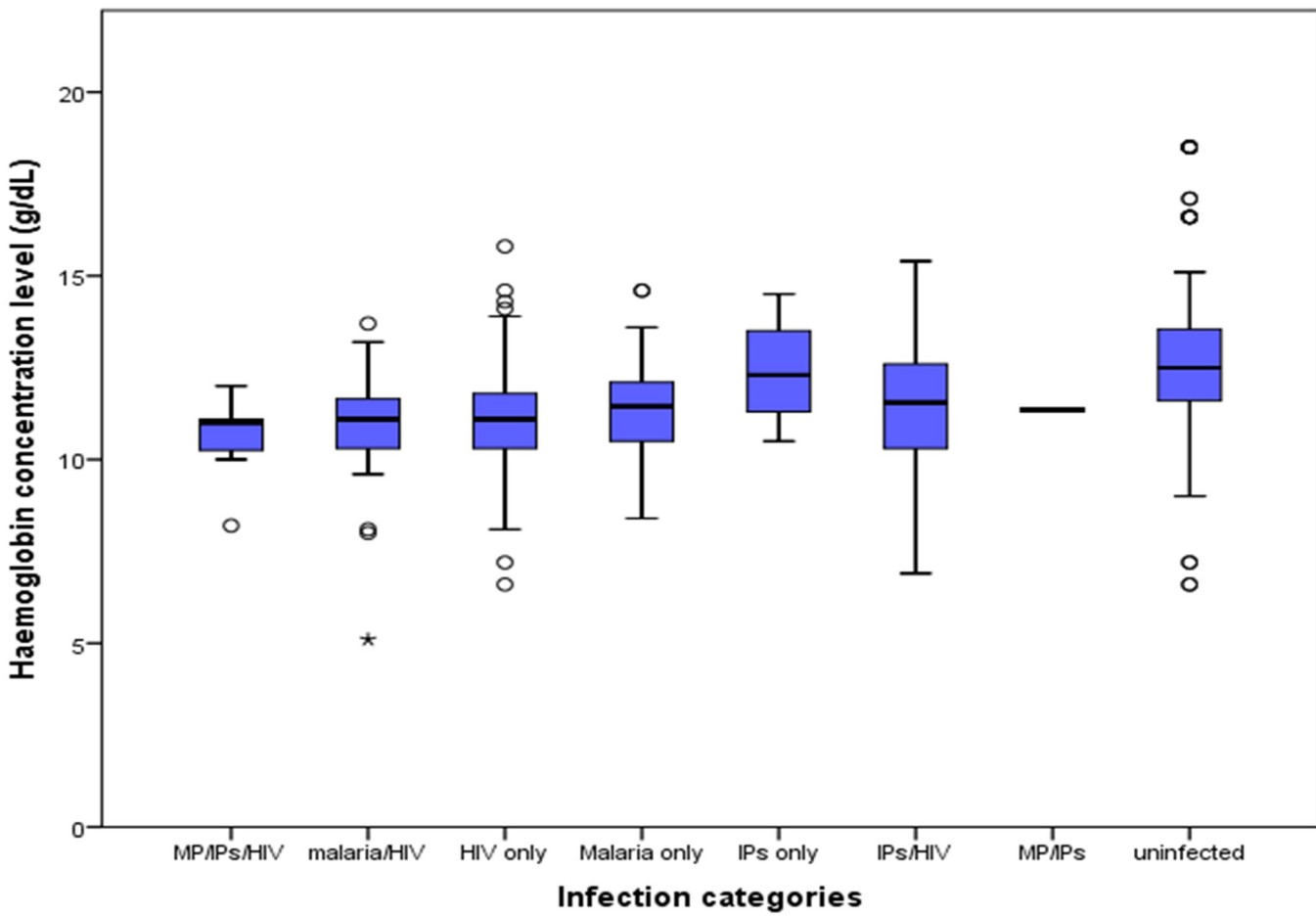

**Fig 3. Box and whisker plots showing mean haemoglobin levels by infection categories.**

### Risk factors of anaemia

A logistic regression model was used to determine the factors associated with anaemia prevalence. The model revealed sex ($P < 0.001$), age group ($P = 0.003$), fever status ($P = 0.035$), infection with HIV only ($P = 0.024$), infection with MP only ($P = 0.004$) as well as HIV/MP co-infection ($P = 0.027$) as significant predictors of anaemia in the study population (Table 4). Male participants and those aged 21–30 years of age were respectively 3 and 2 times more likely to be anaemic than their respective counterparts.

### Discussion

Parasitic infections (especially malaria and intestinal parasites) are major public health concerns among PLWH in developing countries, particularly Sub-Saharan Africa which has been reported to have the highest burden of HIV [2]. This study was a community-based retrospective cohort study carried out to determine the prevalence of malaria and intestinal parasites and its association with anaemia among PLWH and HIV-negative individuals in Buea, in the South West Region of Cameroon.

In the present study, the overall prevalence of malaria parasite was lower than the recently reported national prevalence (24.4%) [33]. This decrease could be attributed to the restless efforts of the Cameroon Government over the years to implement malaria control strategies to

**Table 3. Prevalence of anaemia, anaemia Severity and Mean (SD) Hb levels by infection category.**

| Infection category | Number examined | Prevalence of anaemia | Mean (SD) Hb (g/dL) | Prevalence of anaemia severity | | |
|---|---|---|---|---|---|---|
| | N | n (%) | | Mild anaemia n (%) | Moderate anaemia n (%) | Severe anaemia n (%) |
| PLWH | 190 | 146 (76.8) | 11.11 (1.54) | 108 (56.8) | 35 (18.5) | 3 (1.6) |
| HIV-negative | 216 | 86 (39.8) | 12.45 (2.06) | 72 (33.3) | 12 (5.6) | 2 (0.9) |
| Test statistics | | $\chi^2$ = 56.59, | U = 29316 | | $\chi^2$ = 58.74, | |
| | | P < 0.001* | P < 0.001* | | P < 0.001* | |
| HIV only | 121 | 96 (79.3) | 11.06 (1.44) | 72 (59.5) | 23 (19.0) | 1 (0.8) |
| MP only | 28 | 20 (71.4) | 11.57 (1.52) | 18 (64.3) | 2 (7.1) | 0 (0.0) |
| IPs only | 8 | 2 (25.0) | 12.40 (1.52) | 2 (25.0) | 0 (0.0) | 0 (0.0) |
| Test statistics | | $\chi^2$ = 12.12, | H = 6.87 | | $\chi^2$ = 14.47, | |
| | | P = 0.002* | P < 0.001* | | P = 0.02* | |
| MP/HIV | 24 | 19 (79.2) | 10.83 (1.81) | 15 (62.5) | 3 (12.5) | 1 (4.2) |
| HIV/IPs | 38 | 25 (65.8) | 11.49 (1.68) | 17 (44.7) | 7 (18.4) | 1 (2.6) |
| MP/Ips | 4 | 2 (50.0) | 11.35 (0.06) | 2 (50.0) | 0 (0.0) | 0 (0.0) |
| MP/HIV/IPs | 7 | 6 (85.7) | 10.55 (1.20) | 4 (57.1) | 2 (28.6) | 0 (0.0) |
| Test statistics | | $\chi^2$ = 2.88, | H = 3.84 | | $\chi^2$ = 5.07, | |
| | | P = 0.83 | P = 0.28 | | P = 0.82 | |

H: Kruskal Wallis test; U Mann Whitney U test.

curb down malaria morbidity and mortality on the entire territory; knowing that Cameroon comes fourth on the list of countries that account for the burden of malaria in the WHO African Region [6].

The prevalence of malaria parasite among PLWH in this study was higher compared to the prevalence reported in PLWH in other areas of the South West Region of Cameroon. This includes the 7.8% reported in Limbe [11] and 2.3% in Buea [34] in 2019 and 2018 respectively. It was also observed in this study that despite the efforts made to scale up insecticide-treated nets (ITNs) distribution so that universal coverage can be attained, coverage remains low (57.4%) and this could have contributed to this rise in malaria prevalence among PLWH. Another reason for this difference could be the difference in the age distribution of the study population.

The prevalence of malaria parasite was slightly higher in PLWH than in their HIV-negative counterparts in line with findings in studies carried out in Nigeria [35]. Interactions between malaria and HIV have been studied extensively, which consider PLWH as a high-risk group for malaria in endemic areas like Cameroon [10, 11, 34, 36]. This slight difference is not significant and this could be as a result of the government actions through the availability of different ACTs and ITNs for both PLWH and HIV-negative Cameroonians; as well as the increasing availability and rapid scale-up of ART and cotrimoxazole to PLWH in particular. It is worth noting that the HIV negative participants in the communities with malaria parasite are asymptomatic carriers and could greatly fuel the transmission of the malaria parasite in the community. However, the low prevalence shows that strategies implemented by the Cameroon government over the years are effective as this prevalence is lower than the reported national prevalence of malaria (29%) in 2018 [6].

The febrile status of most of the participants in the study irrespective of their HIV status was attributed to the presence of malaria parasite. This is in line with other studies carried out in Cameroon [11] and in Gabon [37]. Also, feverish PLWH had the highest prevalence of MP/

**Table 4. Logistic regression model examining factors that influence the prevalence of anaemia in the study population.**

| Variable | Bivariate | | Multivariate | |
|---|---|---|---|---|
| | COR | P-value | AOR | P value |
| **Age group (years)** | | | | |
| ≤ 20 | 1.18 | 0.521 | - | - |
| 21–30 | 1.79 | 0.032* | - | - |
| 31–40 | 0.92 | 0.771 | - | - |
| > 40 | Reference | - | - | - |
| **Sex** | | | | |
| Male | 2.68 | < 0.001* | 0.37 | < 0.001* |
| Female | Reference | - | reference | - |
| **Febrile status** | | | | |
| Febrile | 0.52 | 0.035* | 1.92 | 0.010* |
| Non-febrile | Reference | - | reference | - |
| **Single infection** | | | | |
| HIV only | 0.13 | 0.024* | 0.18 | 0.020* |
| MP only | 0.08 | 0.004* | 0.10 | 0.003* |
| IP only | Reference | - | reference | - |
| **Dual infection** | | | | |
| HIV/MP | 0.31 | 0.027* | - | - |
| HIV/IP | 0.63 | 0.192 | - | |
| MP/IP | 1.21 | 0.851 | - | - |
| None | Reference | - | - | - |
| **Triple infection (HIV/MP/IP)** | | | | |
| Yes | 0.30 | 0.282 | - | - |
| No | Reference | - | - | - |
| **Suppressed viral load** | | | | |
| Yes | 1.26 | 0.600 | - | - |
| No | Reference | - | - | - |
| **Zidovudine-based ART** | | | | |
| Yes | 0.78 | 0.695 | - | - |
| No | Reference | - | - | - |
| **CD4 T-cell count (cells/μL)** | | | | |
| < 200 | 0.35 | 0.212 | - | - |
| 200–500 | 0.64 | 0.361 | - | - |
| > 500 | Reference | - | - | - |

*Statistically significant P-value.

The test used in the logistic regression was the Likelihood Ratio test.

IPs co-infection asserting the fact that fever remains an important feature in the clinical diagnosis of infections. Fever is one of the most common clinical signs and characteristic feature of both parasitic and microbial infections.

The prevalence of IPs in Cameroon varied from 27.8% in 2012 [38], to 14.6% in 2013 [39] and 13.0% in this present study. Considering the HIV status, PLWH had a higher prevalence of IPs when compared with their HIV-negative counterparts. This prevalence among PLWH is lower than the 59.52% [39] in 2013 and 57.48% [40] in 2014 reported among PLWH in the Centre and West Regions of Cameroon respectively. This observed decrease in the prevalence of IPs among HIV patients may be accredited to the attainment of viral suppression by

majority of PLWH as well as their better awareness of IPs and their causes. It may also be due to the improvement in the care provided to PLWH by the government of Cameroon by ensuring a constant supply and provision of ART free of charge.

This study revealed a low prevalence of MP and IPs co-infection among PLWH. Limited information is available on co-infection among PLWH in Cameroon; but when compared with other studies carried out among children [19, 41, 42], the prevalence is low. The mean age of PLWH, as well as the undetectable viral load observed in the majority of these patients, could account for this low prevalence. The lower the viral load, the faster the patient's immune system will recover, increasing the chances of fighting any infection easily. Besides, the majority of these patients have been on ART regularly for more than a year which probably could have contributed in boosting their immune systems against these parasites.

PLWH had a higher prevalence of anaemia when compared with their HIV-negative counterparts. This prevalence is higher than that reported in several studies carried out among PLWH [10, 11, 43–47]. The reason for the observed difference might be due to the heterogeneity of the study populations as the previously reported anaemia prevalence values were those of children or adult HIV patients, while this study included both children and adult participants (1–70 years old). Malaria and HIV positivity, as well as fever, were identified in this study as risk factors of anaemia and therefore contributed significantly to the high prevalence of anaemia. Mixed nutritional deficiencies (iron, folic acid, or vitamin $B_{12}$), though not investigated in this study, could have contributed to this high prevalence of anaemia as reported by a study carried out by Volberding *et al.* [48]. A direct comparison of the prevalence of anaemia in different studies is difficult as the study population, inclusion and exclusion criteria and anaemia definitions were different.

Parasites like the malaria parasites and intestinal parasites have long been recognized as major contributors to reduced haemoglobin levels in endemic countries like Cameroon, thereby causing anaemia. Also, PLWH on Zidovudine-containing ART have been reported to influence the haemoglobin level in previous studies [10, 11, 49–51]. In this study, patients infected with HIV, malaria and IPs had a significantly lower mean haemoglobin value. This is in line with previous studies which reported though among children, low haemoglobin values in patients with co-infections when compared with those with single infections [52, 53]. Aside from well-reported side effects of anaemia associated with ZDV-containing ART regimen, though not confirmed in this study, anaemia among HIV infected individuals is more broadly multifactorial in origin, and three basic pathophysiologic mechanisms of anaemia in HIV infection have been postulated [48]: decreased RBC production, increased RBC destruction, and ineffective RBC production.

A high prevalence of severe anaemia was observed among patients co-infected with MP and HIV. This is consistent with studies carried out in Nigeria [54, 55], Ethiopia [49] and more recently in Cameroon [11]. This confirms the possibility that co-infections may increase malaria morbidity, further worsening malaria disease severity [56]. In addition, triple, dual, HIV and malaria parasite infections as well as fever were reported as risk factors of anaemia in this study.

One strength of this study is that it fills some gaps on the occurrence of MP/IPs co-infection among PLWH in the region which had not been reported previously as well as the association between these infections and anaemia among this vulnerable group. Further evidence is demonstrated on the outcome of the efforts in management implemented by the government which are encouraging in reducing the HIV/AIDS morbidity in the various communities. Notwithstanding, this study had as limitations the fact that nutritional deficiency as a cause of anaemia was not investigated. Nevertheless, these findings are valuable to policymakers in guiding the management of these conditions. However, future longitudinal studies on the

impact of iron supplementation and vitamin $B_{12}$ levels among PLWH may be of great relevance to understand the cause of the high anaemia prevalence observed.

## Conclusions

From this study, it can be concluded that malaria parasite and intestinal parasites remain a public health concern among PLWH with higher occurrences of both MP, IPs and MP/IP co-infections than in their HIV negative counterparts in the communities. Though a good majority of the HIV patients in the study had attained viral suppression, they are still more vulnerable to some of the parasites than HIV negative individuals and therefore can constitute a good source of contamination in their various communities. To achieve HIV epidemic control in 2021, the Cameroon Government needs to intensify the implementation of the various strategies. Anaemia stands as a serious haematological abnormality among HIV patients and gets exacerbated even with the viral load suppression in the population of PLWH. Hence, there is a need for constant monitoring of the health status as well as the nutritional status of this at-risk group to better manage them.

## Supporting information

**S1 File.**
(DOCX)

## Acknowledgments

The authors thank all the participants (both children and adults) who took part in the study as well as the staff at the Unit in-charge of patients with HIV (UPEC) in the Buea Regional Hospital. We are also thankful to the Buea community inhabitants who consented and parents who consented for their children to participate in the study.

## Author Contributions

**Conceptualization:** Sorelle Mekachie Sandie.

**Data curation:** Sorelle Mekachie Sandie.

**Formal analysis:** Sorelle Mekachie Sandie, Martin Mih Tasah.

**Investigation:** Sorelle Mekachie Sandie.

**Methodology:** Sorelle Mekachie Sandie, Martin Mih Tasah.

**Project administration:** Sorelle Mekachie Sandie, Irene Ule Ngole Sumbele, Helen Kuokuo Kimbi.

**Software:** Sorelle Mekachie Sandie.

**Supervision:** Irene Ule Ngole Sumbele, Helen Kuokuo Kimbi.

**Validation:** Irene Ule Ngole Sumbele, Helen Kuokuo Kimbi.

**Visualization:** Irene Ule Ngole Sumbele, Helen Kuokuo Kimbi.

**Writing – original draft:** Sorelle Mekachie Sandie.

**Writing – review & editing:** Sorelle Mekachie Sandie.

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
