## [Decision Letter · Decision Letter 0]

28 Jul 2020

PONE-D-20-20708

Malaria and intestinal parasite co-infection and its association with anaemia among people living with HIV in Buea, Southwest Cameroon: A community-based cross-sectional study

PLOS ONE

Dear Dr. Sandie,

Thank you for submitting your manuscript to PLoS ONE. After careful consideration, we felt that your manuscript requires substantial revision, following which it can possibly be reconsidered, thus governing the decision of a “major revision”. As requested by the reviewers, the authors need to address several concerns, particularly related to the data analysis, methods and results. Finally, it is essential to incorporate the limitations of the study otherwise it might compromise data interpretation.   For your guidance, a copy of the reviewers' comments was included below.

We look forward to receiving your revised manuscript.

Kind regards,

Luzia Helena Carvalho, Ph.D.

Academic Editor

PLOS ONE

Journal Requirements:

2. Thank you for indicating in the text of your manuscript that you obtained written informed consent from participants. Please also add this information to your ethics statement in the online submission form.

Reviewers' comments:

Reviewer's Responses to Questions

**Comments to the Author**

1. Is the manuscript technically sound, and do the data support the conclusions?

Reviewer #1: Yes

Reviewer #2: Partly

2. Has the statistical analysis been performed appropriately and rigorously? 

Reviewer #1: Yes

Reviewer #2: No

3. Have the authors made all data underlying the findings in their manuscript fully available?

Reviewer #1: Yes

Reviewer #2: No

4. Is the manuscript presented in an intelligible fashion and written in standard English?

Reviewer #1: Yes

Reviewer #2: Yes

5. Review Comments to the Author

Reviewer #1: The manuscript is well written but the authors need to provide clear description on laboratory methods. The results are well presented but table three need rearrangement to be clear to the reader. The conclusions are based on the study findings

Reviewer #2: Thank you for giving me the opportunity to review this manuscript, which is rather well written and clear. The manuscript adds on the evidence of various conditions common is sub-Saharan Africa and links them to the HIV pandemic.

However, in spite of its merits, the manuscript should undergo some review prior potential publication.

Both major and minor comments are explained below in order of the page of the manuscript:

Page 3 – Lines 52-55: the authors start the manuscript by trying to give evidence of what is known on the issue, which is good. However, this part could be expanded; there is literature that the authors can consider to make their statements stronger (as an example: https://doi.org/10.1016/j.gore.2019.07.002 ). Therefore, I suggest an additional search in the literature.

Page 6 – Lines 126-127: It is not clear how the participants were selected, specifically in terms of their HIV status. Were 2 different selections done on the basis of HIV status, meaning, were the authors specifically going to look for HIV+ and HIV- participants? If so, this study should be redesigned as a (retrospective) cohort since participants are selected on the base of an exposure (i.e. HIV) and see the differences in outcomes (i.e. anaemia, IP, etc.). In cross-sectional studies, a sample of the population is selected, not necessarily on the base of some individual variables. Please, clarify this point and rectify it all along the manuscript.

Page 9 – Lines 194-196: the authors stated that they conducted a binary logistic regression, which appear somehow in contradiction with what written on line 281 (“A multinomial regression analysis was used…”). In logistic regression, usually we talk about “crude” (or univariate) analysis if there is only one exposure and the outcome in the model, or “adjusted” (or multivariable) analysis if there are more than one exposure in the same model. Please, specify which one you used, and if it is crude, please explain (with good reason) why an adjusted analysis was not conducted (which should be). In addition, please, describe which statistical test was used in the logistic regression (which can either be the Wald test or the Likelihood Ratio test).

Page 10-11 (table 1) and page 12-13 (table 2): you provide descriptive statistics for various variables of participants. Please, provide the p-value from the Chi2 test to see if there are statistically significant differences in the various variables between HIV+ and HIV- participants.

Page 16 – table 4: Please, again explain whether the OR are crude or adjusted (see point above); please, include (for clarity) what is the reference for each variable (e.g. if males have a OR of 0.37 for anaemia, it is in comparison to women, which will be the reference) in the table, rather than in the table legenda; please, also include which statistical test was used (see above).

Page 17 onwards (all Discussion section): please reorganize the whole Discussion/Conclusion section. This should be according to a logical sequence. Start by summarizing the findings of your study (no need to repeat values such as percentages and means in the discussion section), how they compare with other similar studies, what are the reasons for differences with other studies. Provide a plausibility for your findings. Provide Strengths and Limitations of your study (this is essential to see to what extent you can give recommendations following your results). Once you have gone through the strengths and limitations of your study, you should provide which are the implications of your results, these may be clinical, public health/programmatic, advocacy etc. as recommendations plus all what you consider to be done as additional research on the topic.

I hope these comments are useful for you.

With my warmest regards.

6. PLOS authors have the option to publish the peer review history of their article (what does this mean?). If published, this will include your full peer review and any attached files.

Reviewer #1: No

Reviewer #2: **Yes: **Rodolfo Rossi

---

## [Author Response · Author response to Decision Letter 0]

26 Aug 2020

Responses to Editor’s Concerns

The manuscript has been corrected to meet PLOS ONE’s style requirements as suggested by the editor.

2. Thank you for indicating in the text of your manuscript that you obtained written informed consent from participants. Please also add this information to your ethics statement in the online submission form.

 Information on ethics statement has been included in the online submission form.

The questionnaire for this study was developed by the principal investigator. A copy has been uploaded as Supporting Information in the online submission.

 

Corrections and Concerns of Reviewer #1

The corrections proposed by reviewer #1 have been effected and are all highlighted in yellow in the revised manuscript.

Reviewer #1: The manuscript is well written but the authors need to provide clear description on laboratory methods. The results are well presented but table three need rearrangement to be clear to the reader. The conclusions are based on the study findings

Thanks for the comments. Laboratory methods, Table 3 in the results section as well as the conclusions as suggested by the reviewer have been upgraded and reorganized.

Abstract

Results 

Line 37: But this is not statistically significant

Effectively, the difference was not statistically significant across the two groups; but as malaria parasite prevalence was one of the main outcomes in the study, it could not be left out in the abstract because of that. Apart from statistical significance, it is also important to know the trend of the various infections across the groups.

Methods

Study design and sample size 

Line 138: This sample size was not intended/powered to compare two proportions

You raised a pertinent issue concerning sample size calculation. To address that, the following sample size formula was used as we are comparing two proportions (See lines 138-143).

N=2(Zα + Zβ)2 P(1-P)/(P0-P1)2 

Where;

N=sample size

Zα= 1.96

Zβ= 0.84

P = Pooled prevalence = (P0 + P1)/2

P0 = 50% (unknown prevalence)

P1 = 67.2% (Chanyalew and Gurara, 2014)

The calculated sample size (N) was 128 for each group.

Sample collection and processing 

Line 164: Briefly describe how you quantified the parasites

A statement on quantification of parasites has been included in the revised manuscript (See lines 164 – 166).

Line 167: Provide details how quantification was done and quality assurance of the readings

A statement on malaria parasite quantification has been included in the revised manuscript (See lines 166 – 172). 

The authors omitted to mention that blood smears were prepared and read by 2 qualified microscopists, and that a third microscopist was requested in case of discrepancy in some results; although that was effectively done. This information has been included under sample collection and processing (See lines 166 -169). 

For quality assurance, a negative result was declared after checking at least 200 high power microscopic fields.

Line 185: Please provide reference

On line 185, the authors just stated the apparatus used for haemoglobin measurement; giving details on the location of the manufacturing company. The authors do not understand which reference should be added on that statement.

Table 3

Line 276: It will be easier to interpret this table if rearranged into two categories HIV+ve and HIV –ve like table2, then see how anemia and co-infection vary across groups.

In Table 3, the authors wanted to bring out clearly the difference in the prevalence of anaemia and its severity with respect to the various categories of infections (single, double and triple infections); so if arranged into two categories, that difference will not be clearly seen.

Figure 1: Please indicate that these participants were excluded/not enrolled.

A statement on that has been included under the description of socio-demographic data and referred to Figure 1 (See lines 216 – 218).

 

Corrections and Concerns of Reviewer #2

The corrections proposed by reviewer #2 (Rodolfo Rossi) have been effected and are all highlighted in turquoise in the revised manuscript.

Reviewer #2: Thank you for giving me the opportunity to review this manuscript, which is rather well written and clear. The manuscript adds on the evidence of various conditions common is sub-Saharan Africa and links them to the HIV pandemic.

However, in spite of its merits, the manuscript should undergo some review prior potential publication.

Both major and minor comments are explained below in order of the page of the manuscript:

Page 3 – Lines 52-55: the authors start the manuscript by trying to give evidence of what is known on the issue, which is good. However, this part could be expanded; there is literature that the authors can consider to make their statements stronger (as an example: https://doi.org/10.1016/j.gore.2019.07.002 ). Therefore, I suggest an additional search in the literature.

Thank you for the suggestion, which has been taken into consideration by the authors. After literature search, the background has been expanded to make the statements stronger (See lines 54 – 62).

Page 6 – Lines 126-127: It is not clear how the participants were selected, specifically in terms of their HIV status. Were 2 different selections done on the basis of HIV status, meaning, were the authors specifically going to look for HIV+ and HIV- participants? If so, this study should be redesigned as a (retrospective) cohort since participants are selected on the base of an exposure (i.e. HIV) and see the differences in outcomes (i.e. anaemia, IP, etc.). In cross-sectional studies, a sample of the population is selected, not necessarily on the base of some individual variables. Please, clarify this point and rectify it all along the manuscript.

The authors believe this study is a cross-sectional study because the participants were recruited from the communities not based on their exposure as well as from the HIV treatment centre in the community to make up the numbers of HIV positive cases. Hence, we had both exposed (HIV patients who already knew their status) and unexposed participants (HIV negatives who had no knowledge of their status and were tested to confirm their HIV status) recruited from both the community and HIV treatment centre. Those who were tested positive for HIV in the community outreach were included in the HIV positive group.

Page 9 – Lines 194-196: the authors stated that they conducted a binary logistic regression, which appear somehow in contradiction with what written on line 281 (“A multinomial regression analysis was used…”). In logistic regression, usually we talk about “crude” (or univariate) analysis if there is only one exposure and the outcome in the model, or “adjusted” (or multivariable) analysis if there are more than one exposure in the same model. Please, specify which one you used, and if it is crude, please explain (with good reason) why an adjusted analysis was not conducted (which should be). In addition, please, describe which statistical test was used in the logistic regression (which can either be the Wald test or the Likelihood Ratio test).

The authors noticed the contradiction raised by the reviewer concerning the analysis that was used to determine odds ratio and it has been addressed in the revised manuscript. 

To address the concern regarding odd ratios, the data was reanalyzed and both the crude and adjusted odds ratios were included in table 4 in the revised manuscript.

The statistical test used was the Likelihood Ratio test.

Page 10-11 (Table 1) and page 12-13 (Table 2): you provide descriptive statistics for various variables of participants. Please, provide the p-value from the Chi2 test to see if there are statistically significant differences in the various variables between HIV+ and HIV- participants.

As seen in Tables 1 and 2, the P- and Chi-square values have been included to show whether there were statistical significance differences in the various variables between HIV positive and negative participants.

Page 16 – Table 4: Please, again explain whether the OR are crude or adjusted (see point above); please, include (for clarity) what is the reference for each variable (e.g. if males have a OR of 0.37 for anaemia, it is in comparison to women, which will be the reference) in the table, rather than in the table legenda; please, also include which statistical test was used (see above).

The reference for each variable has been included in table 4 for better understanding of the results (See line 308).

Page 17 onwards (all Discussion section): please reorganize the whole Discussion/Conclusion section. This should be according to a logical sequence. Start by summarizing the findings of your study (no need to repeat values such as percentages and means in the discussion section), how they compare with other similar studies, what are the reasons for differences with other studies. Provide a plausibility for your findings. Provide Strengths and Limitations of your study (this is essential to see to what extent you can give recommendations following your results). Once you have gone through the strengths and limitations of your study, you should provide which are the implications of your results, these may be clinical, public health/programmatic, advocacy etc. as recommendations plus all what you consider to be done as additional research on the topic.

The discussion section has been improved as requested by the reviewer (See lines 423-433).

---

## [Decision Letter · Decision Letter 1]

29 Sep 2020

PONE-D-20-20708R1

Malaria and intestinal parasite co-infection and its association with anaemia among people living with HIV in Buea, Southwest Cameroon: A community-based cross-sectional study

PLOS ONE

Dear Dr. Sandie,

Thank you for submitting your manuscript for review to PLoS ONE. After careful consideration, we feel that your manuscript will likely be suitable for publication if it is revised to address specific points raised by the reviewers. A significant number of topics still need to be clarified and manuscript should be adjusted as suggested.   A major concern was related to study design (for example, criteria of recruitment) that should be revised as requested.  For your guidance, a copy of the reviewer' comments was included below. 

We look forward to receiving your revised manuscript.

Kind regards,

Luzia Helena Carvalho, Ph.D.

Academic Editor

PLOS ONE

Reviewers' comments:

Reviewer's Responses to Questions

**Comments to the Author**

1. If the authors have adequately addressed your comments raised in a previous round of review and you feel that this manuscript is now acceptable for publication, you may indicate that here to bypass the “Comments to the Author” section, enter your conflict of interest statement in the “Confidential to Editor” section, and submit your "Accept" recommendation.

Reviewer #1: (No Response)

Reviewer #2: (No Response)

2. Is the manuscript technically sound, and do the data support the conclusions?

Reviewer #1: No

Reviewer #2: Yes

3. Has the statistical analysis been performed appropriately and rigorously? 

Reviewer #1: No

Reviewer #2: Yes

4. Have the authors made all data underlying the findings in their manuscript fully available?

Reviewer #1: No

Reviewer #2: Yes

5. Is the manuscript presented in an intelligible fashion and written in standard English?

Reviewer #1: No

Reviewer #2: Yes

6. Review Comments to the Author

Reviewer #1: Major issues with this manuscript

is in the study design (cross sectional study) but sampling of participants in the clinics (PLWH) and community (HIV -ve participants) is not well described.

In the analysis the authors are not only comparing the two groups PLWH and HIV -ve they also add other sub groups which makes to interpret the findings.

Throughout the text they use either PLHW or HIV +ve to refer to the same group of participants

Reviewer #2: Dear authors,

Thank you for addressing the vast majority of the comments I had made during the first revision.

There are still 2 pending issues that could be better addressed

1) lines 130 131 "The two categories of participants, PLWH and HIV-negative individuals were enrolled in the study at the HIV Treatment and Care Centre and from the different neighbourhoods"

By reading this, I still understand that participants were recruited according to their HIV status. Therefore I kindly ask you to reformulate since you state that your study is a cross-sectional study

2) In the discussion section, when summarizing the results, I suggest removing all/most numerical values since these are already reported in the results section.

Thank you again for your efforts. I wish you a good continuation on your work

7. PLOS authors have the option to publish the peer review history of their article (what does this mean?). If published, this will include your full peer review and any attached files.

Reviewer #1: No

Reviewer #2: **Yes: **Rodolfo Rossi

---

## [Author Response · Author response to Decision Letter 1]

17 Oct 2020

Responses to Reviewers Comments and Concerns

We are submitting here responses to the concerns raised by reviewers on manuscript: PONE-D-20-20708.

Title:

Malaria and intestinal parasite co-infection and its association with anaemia among people living with HIV in Buea, Southwest Cameroon: A community-based cross-sectional study

Authors

Sorelle Mekachie Sandie*: sandiesorelle2@gmail.com

Irene Ule Ngole Sumbele: sumbelei@yahoo.co.uk

Martin Mih Tasah: tasahmartins@gmail.com

Helen Kuokuo Kimbi: hkimbi@yahoo.co.uk

Version: 3

Date: 17th October 2020

Authors’ responses to reviewers’ concerns over leaf.

 

Corrections and Concerns of Reviewer #1

The corrections proposed by reviewer #1 have been effected and are all highlighted in yellow in the revised manuscript.

Is in the study design (cross sectional study) but sampling of participants in the clinics (PLWH) and community (HIV -ve participants) is not well described.

The sampling of participants for both categories has been described in the study design section as suggested (See lines 137 – 144).

In the analysis the authors are not only comparing the two groups PLWH and HIV -ve they also add other sub groups which makes to interpret the findings. Throughout the text they use either PLHW or HIV +ve to refer to the same group of participants

In this study, PLWH and HIV positive patients referred to the same group of participants. For better clarification and interpretation, the term “PLWH” has been used throughout the manuscript instead of HIV positive patients. 

Abstract

Results 

Line 35: Here you could have a sentence to show the prevalence results; PLWH vs HIV –ve instead of overall prevalence in the first sentence.

The authors are grateful for the suggestion. However, the authors believe it is most appropriate to give the prevalence of the different pathogens under study before stating the prevalence in association with different conditions examined in the study for a logical presentation. This presentation ties with the title of the manuscript. (See lines 35-39). 

Line 38: Use the same term PLWH and show mean Hb for the 2 groups.

The term PLWH has been used in Line 38 and the mean Hb for the 2 groups has been reported (See lines 38-39).

Line 40: This was not the focus of your study.

The statement of line 40 has been removed as it was not the focus of the study.

Line 41: Use PLWH instead of HIV positive patients.

The term “HIV positive patients” has been replaced by “PLWH” throughout the manuscript.

Methods

Study design and sample size 

Line 137: This study also collected data from clinics as stated above in the description of study area.

Yes. That has been clearly stated in lines 140-143 and lines 153-155. PLWH recruited from the HIV Treatment and Care centre of the hospital had data from the questionnaire verified with that in their file in the centre. In addition, the appropriate laboratory data measured on the day of recruitment was also obtained from the treatment centre. Hence, the description of data collection at the level of the HIV Treatment and Care Centre has been included in this section as mentioned in the lines above.

Line 138: It is not clearly stated how the HIV positive individuals were sampled in the community or health facility.

A more comprehensive statement on how HIV positive patients were recruited in this study can be found in the study design section (See lines 141-145).

Line 139: Why convenient sampling?

The sampling method used was a purposive convenient sampling method. Purposive because the study was meant to compare two groups of participants and convenient because the participants for the study could be easily enrolled from this cohort of individuals. It was quite convenient for them to be enrolled as they presented themselves and gave their consents (See line 137).

Table 2

Line 252: How many HIV positive patients were screened in the community?

From the outreach screening programme in the community, four participants were found living with HIV that were referred to the HIV-care and treatment centre for their enrolment follow-up and collection of appropriate data.

Malaria parasite, Intestinal parasite infections and malaria/intestinal parasite co-infection

Line 257: Decide which term to use PLWH or HIV positive participants.

PLWH has been used throughout the manuscript.

Line 258: Recalculate this is around 16.6 65/406.

Effectively there was a problem with the calculation of the prevalence of IPs in the total population. PLWH infected with IPs were 45 in number giving a prevalence of 23.7% while HIV negative participants infected with IPs were 12 in number giving a prevalence of 5.6%. The total number of participants infected with IPs irrespective of their HIV status was supposed to be 45 + 12 = 57, giving a total prevalence of 14.0%. The correction has been effected in the manuscript (See line 259-260) as well as in Figure 2.

Line 263: This sentence should be rephrased, although not statistically significant, malaria/IP co-infection was higher in PLWH compared to HIV –ve participants.

The sentence on line 263 has been rephrased appropriately.

Corrections and Concerns of Reviewer #2

The corrections proposed by reviewer #2 (Rodolfo Rossi) have been effected and are all highlighted in turquoise in the revised manuscript.

Thank you for addressing the vast majority of the comments I had made during the first revision.

There are still 2 pending issues that could be better addressed

Lines 130 – 131: "The two categories of participants, PLWH and HIV-negative individuals were enrolled in the study at the HIV Treatment and Care Centre and from the different neighbourhoods". By reading this, I still understand that participants were recruited according to their HIV status. Therefore I kindly ask you to reformulate since you state that your study is a cross-sectional study

The authors believe this study is not a retrospective cohort study because both the exposure (HIV status) and outcome (Malaria, anaemia, IPs) were measured at the same time, making it a cross-sectional study. Hence, participants were recruited in a cross-sectional study and grouped according to their status into PLWH and HIV- negative participants. The statement has been reformulated as suggested by the reviewer for better understanding of the study design (See lines 129-130)

Discussion

In the discussion section, when summarizing the results, I suggest removing all/most numerical values since these are already reported in the results section.

As suggested by the reviewer, most numerical values have been removed from the discussion section.

---

## [Decision Letter · Decision Letter 2]

26 Oct 2020

PONE-D-20-20708R2

Malaria and intestinal parasite co-infection and its association with anaemia among people living with HIV in Buea, Southwest Cameroon: A community-based cross-sectional study

PLOS ONE

Dear Dr.  Sandie,

According to the reviewer, the concerns raised during the peer review process were not properly addressed. For example, reviewer #2 complains that the study design is still confuse. According to the reviewer, people HIV+/HIV-  were categorized by exposure at the beginning of the study, therefore the study, as presented, is not a cross-sectional study. At this time we strongly suggest the authors to proper address the concerns related to study design. 

Please submit your revised manuscript by November 20 If you will need more time than this to complete your revisions, please reply to this message or contact the journal office at plosone@plos.org. Please include the following items when submitting your revised manuscript:

We look forward to receiving your revised manuscript.

Kind regards,

Luzia Helena Carvalho, Ph.D.

Academic Editor

PLOS ONE

Reviewers' comments:

Reviewer's Responses to Questions

**Comments to the Author**

1. If the authors have adequately addressed your comments raised in a previous round of review and you feel that this manuscript is now acceptable for publication, you may indicate that here to bypass the “Comments to the Author” section, enter your conflict of interest statement in the “Confidential to Editor” section, and submit your "Accept" recommendation.

Reviewer #1: All comments have been addressed

Reviewer #2: (No Response)

2. Is the manuscript technically sound, and do the data support the conclusions?

Reviewer #1: Yes

Reviewer #2: Partly

3. Has the statistical analysis been performed appropriately and rigorously? 

Reviewer #1: Yes

Reviewer #2: No

4. Have the authors made all data underlying the findings in their manuscript fully available?

Reviewer #1: No

Reviewer #2: Yes

5. Is the manuscript presented in an intelligible fashion and written in standard English?

Reviewer #1: Yes

Reviewer #2: Yes

6. Review Comments to the Author

Reviewer #1: The conclusion in tne abstract should be similar to conclusion at the end of discussion section. ie focusing on burden of infections MP, IPs and Co infection MP/IPs . Currently Conclusion: it is stated Multiple infections exacerbate the health condition of PLWH.

Reviewer #2: Dear authors,

thank you for your efforts, but the issue of study design highlighted by me, the other reviewer and the editor in the previous round of review has not been properly addressed, leaving confusion on the understanding of study design. Even if recruited simultaneously, people HIV+ and HIV- are categorized by exposure at the beginning of the study, therefore the study, as presented is not a cross-sectional study.

I am afraid I will not be able to further review future version of you manuscript.

7. PLOS authors have the option to publish the peer review history of their article (what does this mean?). If published, this will include your full peer review and any attached files.

Reviewer #1: No

Reviewer #2: **Yes: **Rodolfo Rossi

---

## [Author Response · Author response to Decision Letter 2]

28 Oct 2020

Corrections and Concerns of Editor

According to the reviewer, the concerns raised during the peer review process were not properly addressed. For example, reviewer #2 complains that the study design is still confuse. According to the reviewer, people HIV+/HIV- were categorized by exposure at the beginning of the study, therefore the study, as presented, is not a cross-sectional study. At this time we strongly suggest the authors to proper address the concerns related to study design. 

After thorough literature search on research methodology for better understanding, the authors have acknowledged the concerns of the second reviewer on the study design and have addressed them appropriately throughout the manuscript. 

The study design has been modified to a retrospective cohort study design as the exposure (i.e HIV status) was known at the start of the study; making it not a cross-sectional study.

Corrections and Concerns of Reviewer #1

The corrections proposed by reviewer #1 have been effected and are all highlighted in yellow in the revised manuscript.

The conclusion in the abstract should be similar to conclusion at the end of discussion section. ie focusing on burden of infections MP, IPs and Co infection MP/IPs . Currently Conclusion: it is stated Multiple infections exacerbate the health condition of PLWH.

The concern raised concerning the abstract’s conclusion has been addressed appropriately as suggested by the reviewer (See line 46).

Corrections and Concerns of Reviewer #2

The corrections proposed by reviewer #2 (Rodolfo Rossi) have been effected and are all highlighted in turquoise in the revised manuscript.

Thank you for your efforts, but the issue of study design highlighted by me, the other reviewer and the editor in the previous round of review has not been properly addressed, leaving confusion on the understanding of study design. Even if recruited simultaneously, people HIV+ and HIV- are categorized by exposure at the beginning of the study, therefore the study, as presented is not a cross-sectional study.

After thorough literature search on research methodology for better understanding, the authors have acknowledged the concerns on the study design and have addressed them appropriately throughout the manuscript. 

The study design has been modified to a retrospective cohort study design as the exposure (i.e HIV status) was known at the start of the study; not making it a cross-sectional study.

---

## [Decision Letter · Decision Letter 3]

16 Nov 2020

PONE-D-20-20708R3

Malaria and intestinal parasite co-infection and its association with anaemia among people living with HIV in Buea, Southwest Cameroon: A community-based retrospective cohort study

PLOS ONE

Dear Dr. Sandie,

Thank you for submitting your manuscript for review to PLoS ONE. After careful consideration, we feel that your manuscript will likely be suitable for publication if the authors revise it to address critical points raised by the reviewers.  According to reviewers, there are some specific areas where further improvements would be of substantial benefit to the readers, including methods and results. For your guidance, a copy of the reviewers' comments is included below. 

We look forward to receiving your revised manuscript.

Kind regards,

Luzia Helena Carvalho, Ph.D.

Academic Editor

PLOS ONE

Reviewers' comments:

Reviewer's Responses to Questions

**Comments to the Author**

1. If the authors have adequately addressed your comments raised in a previous round of review and you feel that this manuscript is now acceptable for publication, you may indicate that here to bypass the “Comments to the Author” section, enter your conflict of interest statement in the “Confidential to Editor” section, and submit your "Accept" recommendation.

Reviewer #3: (No Response)

2. Is the manuscript technically sound, and do the data support the conclusions?

Reviewer #3: Yes

3. Has the statistical analysis been performed appropriately and rigorously? 

Reviewer #3: Yes

4. Have the authors made all data underlying the findings in their manuscript fully available?

Reviewer #3: Yes

5. Is the manuscript presented in an intelligible fashion and written in standard English?

Reviewer #3: Yes

6. Review Comments to the Author

Reviewer #3: In the study titled “Malaria and intestinal parasite co-infection and its association with anaemia among people living with HIV in Buea, Southwest Cameroon: A community-based retrospective cohort study”, Sorelle et al. determined the prevalence of malaria parasite and intestinal parasites in PLWH compared to HIV negative counterparts and also assessed the effects of these infections on anaemia to provide relevant information to policymakers in the country.

Concerns raised by previous reviewers (about study design and conclusion) have been addressed by the authors. However, to further improve the quality of the manuscript for publication, I suggest that authors revise the manuscript to include the minor but relevant observations below:

Pg 3, lines 54-55: “In 2018, there were... “. What is the current status for new cases (in 2020)? Is there any improvement? Please include this with citations. Also include the current status in Cameroun (line 56).

Pg 3, line 65: “variety of microbial and parasitic infections such as Plasmodium, Mycobacterium tuberculosis... “. I suggest that authors write the infections (malaria, tuberculosis) instead of the microbes, to eliminate any possible ambiguity.

Pg 4, 72-73: “African countries still bear the largest burden... “. 2018 is long time to make this conclusive statement. The situation may have changed within 2 years. I suggest authors provide a recent data/citation to back up this statement.

Pg 6, lines 123-125: “ In the national HIV prevalence ranking... “. The reference (25) show that this impact assessment was done in 2017, hence the South West Region may no longer be the 6th on the ranking in 2020. I suggest that authors obtain more recent data OR rephrase the statement to indicate that information presented is not recent, to avoid misleading prospective readers.

Pg 6, 129-130: There is also inadequate access to safe water... “. Please provide a reference for this.

Study area and participants: age range range of participants was not specified.

Pg 6, 133: “after confirmation of their HIV status ...”. Please specify the steps/assays used for the confirmation of HIV Status.

Pg 7, 143-148: “HIV-negative participants included persons who ... “. This section is describing the study participants. Please move to appropriate section.

Pg 11, 238 “were students (56.5%)...”. I presume authors are referring to the employment status of this category? If so, I suggest the use of ”unemployed” instead if this is the case, as some students engage in work study.

Tables 1 and 2 have grid lines while tables 3 and 4 do not. I suggest a uniformity in table presentation according to journal’s guidelines.

Pg 15: 273-276: “Out of the 406 participants, 1.7% (7) had triple infection...”. Are the authors describing those with triple infection etc from the population or individuals with triple infection etc who were anaemic? If it is haemoglobin level that is being described, anaemia was not indicated anywhere in this section aside from the subtitle. Authors should endeavour to clarify this section.

7. PLOS authors have the option to publish the peer review history of their article (what does this mean?). If published, this will include your full peer review and any attached files.

Reviewer #3: **Yes: **Nneoma Confidence JeanStephanie ANYANWU

---

## [Author Response · Author response to Decision Letter 3]

20 Nov 2020

Corrections and concerns of Reviewer #3: 

Pg 3, lines 54-55: “In 2018, there were... “. What is the current status for new cases (in 2020)? Is there any improvement? Please include this with citations. Also include the current status in Cameroun (line 56).

A more current status of HIV new cases has been given as reported by UNAIDS in 2019 and that report carries information related to the year 2018. That is why the authors specified that this data was for the year 2018; though it was reported by UNAIDS in 2019. With regards to Cameroon, the most current report is that of 2018 as of now as reported by UNAIDS.

Pg 3, line 65: “variety of microbial and parasitic infections such as Plasmodium, Mycobacterium tuberculosis... “. I suggest that authors write the infections (malaria, tuberculosis) instead of the microbes, to eliminate any possible ambiguity.

The sentence has been adjusted to eliminate ambiguity (See line 65). Malaria is the disease that results from an infection with Plasmodium likewise tuberculosis arises because of an infection with M. tuberculosis. Hence, it is necessary to state the infectious agent.

Pg 4, 72-73: “African countries still bear the largest burden... “. 2018 is long time to make this conclusive statement. The situation may have changed within 2 years. I suggest authors provide a recent data/citation to back up this statement.

The most recent published malaria report by WHO is that of 2019, which reports the burden of malaria in 2018. Malaria burden for 2019 has not yet been published by WHO. 

Pg 6, lines 123-125: “ In the national HIV prevalence ranking... “. The reference (25) show that this impact assessment was done in 2017, hence the South West Region may no longer be the 6th on the ranking in 2020. I suggest that authors obtain more recent data OR rephrase the statement to indicate that information presented is not recent, to avoid misleading prospective readers.

The data presented is the most recent published information on HIV in Cameroon. The authors believe that as long as there are no updated reports on the ranking, the South West Region still remains in the 6th position on the ranking. 

Pg 6, 129-130: There is also inadequate access to safe water... “. Please provide a reference for this.

Reference 27 has been added to support the statement and the sentence has been rephrased appropriately to avoid misunderstanding (See line 131 – 133).

Study area and participants: age range of participants was not specified.

As suggested by the reviewer, age range has been added to the statement (See line 135).

Pg 6, 133: “after confirmation of their HIV status ...”. Please specify the steps/assays used for the confirmation of HIV Status.

The test used for HIV confirmation was Determine Test strip and this was specified in the statement (See line 136).

Pg 7, 143-148: “HIV-negative participants included persons who ... “. This section is describing the study participants. Please move to appropriate section.

The section describing the study participants has been moved to the appropriate section as suggested by the reviewer (See lines 136 – 141).

Pg 11, 238 “were students (56.5%)...”. I presume authors are referring to the employment status of this category? If so, I suggest the use of ”unemployed” instead if this is the case, as some students engage in work study.

In our setting in Cameroon, most students are not engaged in work activities and that is why this terms was used for those participants who are students and do not have any lucrative activity. There is a section under employment status for those who are not students and are unemployed and that category is referred to as “unemployment”..

Tables 1 and 2 have grid lines while tables 3 and 4 do not. I suggest a uniformity in table presentation according to journal’s guidelines.

Grid lines have been added to tables 3 and 4 to follow the journal’s guidelines for presentation of tables.

Pg 15: 273-276: “Out of the 406 participants, 1.7% (7) had triple infection...”. Are the authors describing those with triple infection etc from the population or individuals with triple infection etc who were anaemic? If it is haemoglobin level that is being described, anaemia was not indicated anywhere in this section aside from the subtitle. Authors should endeavour to clarify this section.

Those with triple infection are those who were found to be infected with HIV, malaria parasite and intestinal parasites simultaneously. 

Under the section “Infection categories, haemoglobin level and anaemia prevalence”, the first part describes the different infection categories followed by their effects on the haemoglobin level. Then, after figure 3 which displays the differences in mean haemoglobin level with respect to infection categories, anaemia prevalence is discussed and it still falls under the same section.

---

## [Decision Letter · Decision Letter 4]

24 Nov 2020

PONE-D-20-20708R4

Malaria and intestinal parasite co-infection and its association with anaemia among people living with HIV in Buea, Southwest Cameroon: A community-based retrospective cohort study

PLOS ONE

Dear Dr.  Sandie,

Thank you for resubmitting your manuscript to PLoS ONE. Although the data from this study has potential to be informative, relevant topics raised by the reviewer #3 during the peer review process remain to be addressed by the authors. At this time, we strongly suggest the authors to proper address all topics raised by the reviewer.  For your guidance, a copy of the reviewer’s comments was included below.  

We look forward to receiving your revised manuscript.

Kind regards,

Luzia Helena Carvalho, Ph.D.

Academic Editor

PLOS ONE

Reviewers' comments:

Reviewer's Responses to Questions

**Comments to the Author**

1. If the authors have adequately addressed your comments raised in a previous round of review and you feel that this manuscript is now acceptable for publication, you may indicate that here to bypass the “Comments to the Author” section, enter your conflict of interest statement in the “Confidential to Editor” section, and submit your "Accept" recommendation.

Reviewer #3: (No Response)

2. Is the manuscript technically sound, and do the data support the conclusions?

Reviewer #3: Yes

3. Has the statistical analysis been performed appropriately and rigorously? 

Reviewer #3: Yes

4. Have the authors made all data underlying the findings in their manuscript fully available?

Reviewer #3: (No Response)

5. Is the manuscript presented in an intelligible fashion and written in standard English?

Reviewer #3: Yes

6. Review Comments to the Author

Reviewer #3: Authors did not satisfactorily attend to comments and concerns raised in the course of my previous review of the manuscript. It is imperative that the manuscript be revised to address these comments before it can be accepted for publication. Below are the comments:

Pg 3, 65-67: This statement is incorrect. I pointed this out in my previous review. Authors stated “microbial and parasitic infections such as…”. It is expected that the examples to be given would be microbial and parasitic infections, and not their aetiological agents. Plasmodium is not an infection but an agent, likewise Mycobacterium. If the authors insist on mentioning the aetiology of the infections, then I suggest “microbial and parasitic infections… “ be corrected to “microbes such as Plasmodium…”. Authors need not repeat parasites in this revision as microbes adequately captures this in the context.

Pg 4, line 74-75: “African countries still bear the largest… “. Contrary to the authors’ reply to my previous comment, the most recent WHO world malaria report was published in April 2020. Please update information provided in this section.

Pg 6, line 127: Contrary to the authors’ reply to my comment, this assessment was concluded and announced in 2018. Also, summary sheet for this report is dated February 2020 and South West Region is not the 6th but the 7th according to the summary sheet. Please visit https://phia.icap.columbia.edu/cameroon-summary-sheet/ to obtain this resource and revise this section of the manuscript accordingly, as well as the affected reference.

Pg 6, line 136: Authors were required to specify the steps and assays used for the confirmation of HIV Status. This has not been done. Determine is not a confirmation assay, although it is employed as a national algorithm in certain countries alongside others serial assays for diagnostic purposes. Western blot and PCR are confirmatory assays used for confirmation of HIV infection. Did authors employ any of these in confirming the HIV Status of study participants? If yes, please give a detailed description of the assay methods employed under a subtitle.

7. PLOS authors have the option to publish the peer review history of their article (what does this mean?). If published, this will include your full peer review and any attached files.

Reviewer #3: **Yes: **Nneoma Confidence Jeanstephanie Anyanwu

---

## [Author Response · Author response to Decision Letter 4]

25 Nov 2020

Corrections and concerns of Reviewer #3: 

Pg 3, 65-67: This statement is incorrect. I pointed this out in my previous review. Authors stated “microbial and parasitic infections such as…”. It is expected that the examples to be given would be microbial and parasitic infections, and not their aetiological agents. Plasmodium is not an infection but an agent, likewise Mycobacterium. If the authors insist on mentioning the aetiology of the infections, then I suggest “microbial and parasitic infections… “ be corrected to “microbes such as Plasmodium…”. Authors need not repeat parasites in this revision as microbes adequately captures this in the context.

As suggested by the reviewer, the words “microbial and parasitic infections” have been replaced by “microbes such as Plasmodium, Mycobacterium and several intestinal parasites” to eliminate ambiguity (See lines 65-67).

Pg 4, line 74-75: “African countries still bear the largest… “. Contrary to the authors’ reply to my previous comment, the most recent WHO world malaria report was published in April 2020. Please update information provided in this section.

We thank the reviewer for the updated information. The information has been updated accordingly as suggested by the reviewer indicating that the most recently published malaria report is that of 2019 as published in April 2020 (See lines 73-74).

Pg 6, line 127: Contrary to the authors’ reply to my comment, this assessment was concluded and announced in 2018. Also, summary sheet for this report is dated February 2020 and South West Region is not the 6th but the 7th according to the summary sheet. Please visit https://phia.icap.columbia.edu/cameroon-summary-sheet/ to obtain this resource and revise this section of the manuscript accordingly, as well as the affected reference.

We are grateful to the reviewer for providing the link with the most recent information on the position of the South West Region in Cameroon with respect to HIV ranking. The information has been updated r (See lines 125-126).

Pg 6, line 136: Authors were required to specify the steps and assays used for the confirmation of HIV Status. This has not been done. Determine is not a confirmation assay, although it is employed as a national algorithm in certain countries alongside others serial assays for diagnostic purposes. Western blot and PCR are confirmatory assays used for confirmation of HIV infection. Did authors employ any of these in confirming the HIV Status of study participants? If yes, please give a detailed description of the assay methods employed under a subtitle

Due to limited finances, no HIV confirmatory test was carried out in this study and the HIV positivity of the participants was based on positive results obtained using the Determine Test strip. The sentence has been rephrased for better understanding (See line 135).

---

## [Decision Letter · Decision Letter 5]

3 Dec 2020

PONE-D-20-20708R5

Malaria and intestinal parasite co-infection and its association with anaemia among people living with HIV in Buea, Southwest Cameroon: A community-based retrospective cohort study

PLOS ONE

Dear Dr. Sandie,

Thank you for submitting your manuscript to PLoS ONE. After careful consideration, we felt that your manuscript still requires substantial revision, following which, it can possibly be reconsidered, thus governing the decision of a “major revision”. According to the reviewer, the methodology is heavily flawed due to the lack of confirmation of the HIV Status of the study participants. Where confirmation assays are not obtainable for one reason or the other, national algorithm is often employed and acceptable for research purposes. The reviewer strongly suggests the authors to adjust MS’s results otherwise the quality of the MS may be compromised.

We look forward to receiving your revised manuscript.

Kind regards,

Luzia Helena Carvalho, Ph.D.

Academic Editor

PLOS ONE

Reviewers' comments:

Reviewer's Responses to Questions

**Comments to the Author**

1. If the authors have adequately addressed your comments raised in a previous round of review and you feel that this manuscript is now acceptable for publication, you may indicate that here to bypass the “Comments to the Author” section, enter your conflict of interest statement in the “Confidential to Editor” section, and submit your "Accept" recommendation.

Reviewer #3: (No Response)

2. Is the manuscript technically sound, and do the data support the conclusions?

Reviewer #3: Partly

3. Has the statistical analysis been performed appropriately and rigorously? 

Reviewer #3: Yes

4. Have the authors made all data underlying the findings in their manuscript fully available?

Reviewer #3: (No Response)

5. Is the manuscript presented in an intelligible fashion and written in standard English?

Reviewer #3: Yes

6. Review Comments to the Author

Reviewer #3: Authors have done a commendable job in addressing my review comments. However, the methodology is heavily flawed due to the lack of confirmation of the HIV Status of the study participants. Where confirmation assays are not obtainable for one reason or the other, national algorithm is often employed and acceptable for research purposes. National algorithm usually involves serial testing (more than 1 test). Cameroon employs 2 RDTs in 2 or 3 serial tests, depending on whether Indeterminate results were obtained. With only one test (Determine) carried out, it is impossible to tell if there were any false positives/negatives obtained. I suggest that authors visit https://bmjopen.bmj.com/content/8/3/e020611 to study the algorithm used and carry out the same methods in the present study to improve the quality of this study.

Also, I suggest that the inability to carry out further confirmation assays be included in a section with the subtitle "Study limitations". Other challenges/limitations encountered in the course of the study can also be included in this section.

7. PLOS authors have the option to publish the peer review history of their article (what does this mean?). If published, this will include your full peer review and any attached files.

Reviewer #3: **Yes: **Nneoma Confidence JeanStephanie ANYANWU

---

## [Author Response · Author response to Decision Letter 5]

31 Dec 2020

Concerns or comments of the Editor

The reviewer strongly suggests the authors to adjust MS’s results otherwise the quality of the MS may be compromised.

We the authors believe the results are presented in an appropriate manner to address the objective of the study which is to “determine the prevalence of malaria parasite and intestinal parasites in PLWH compared to HIV negative counterparts as well as to assess the effects of these infections on anaemia to provide relevant information to policymakers in the country’. The PLWH were recruited from HIV care and treatment center where testing is usually done following the National algorithm and the HIV negative individuals were recruited from the community upon confirmation of their HIV-negative status. This is the way the results have been presented following the input by the several reviewers. Hence, we the authors believe there is no need for further modification of the presentation of the results at this point since no further study was carried out or the objective of the study has not been changed or rephrased.

Corrections and concerns of Reviewer #3: 

Authors have done a commendable job in addressing my review comments. However, the methodology is heavily flawed due to the lack of confirmation of the HIV Status of the study participants. 

We the authors wish to vehemently reject the appraisal of “methodology is heavily flawed due to lack of confirmation of HIV status”. As stated in the methodology, PLWH were recruited from the HIV treatment and care centers where potential participants in the study were already enrolled members with the centers based on the fact that their HIV status had been confirmed based on the National algorithm in all the treatment and care centers in the national territory. Because of their HIV status, viral load and CD4 counts were regularly evaluated and it was confirmed that they were on anti-retrovirals for varying periods as indicated in the data. Hence, they were confirmed as PLWH. That is why the study design is a retrospective cohort study. For the healthy individuals recruited during the outreach program in the community, their HIV negative status was confirmed using the Determine 1 test which is actually stated in the algorithm that it is highly sensitive and used to confirm HIV negative cases (who lack the corresponding antibodies being detected). 

Where confirmation assays are not obtainable for one reason or the other, national algorithm is often employed and acceptable for research purposes.

The status of all the participants were confirmed, that is why they were grouped into PLWH and HIV negative individuals. The four participants identified in the outreach program that were HIV positive for the Determine1, their confirmation test was obtained from the HIV treatment and care center following their enrolment based on the referral. We will wish to inquire if the reviewer is trying to introduce a new protocol by further confirmation of HIV negative test results of healthy individuals tested by Determine1 which is the standard National algorithm for those who are HIV negative?

 National algorithm usually involves serial testing (more than 1 test). Cameroon employs 2 RDTs in 2 or 3 serial tests, depending on whether Indeterminate results were obtained. With only one test (Determine) carried out, it is impossible to tell if there were any false positives/negatives obtained. I suggest that authors visit https://bmjopen.bmj.com/content/8/3/e020611 to study the algorithm used and carry out the same methods in the present study to improve the quality of this study.

We appreciate the fact that the reviewer referred us to this paper which we had consulted before designing and carrying out the study. It is based on the algorithms that PLWH were enrolled into the study from the HIV care and treatment center where the protocols are strictly followed before a patient is enrolled in the center. 

The present study included both people living with HIV and HIV negative participants from the community. As specified in the methodology, HIV positive participants were recruited from the HIV Treatment and Care center of the hospital and therefore their positive HIV status had already been confirmed as prescribed by the Cameroon national algorithm i.e using Determine as the first line test and using Oraquick as the confirmatory test before placing them on ART. Determine 1 test was used in this study to confirm the negative HIV status of the HIV negative participants from the community before enrolment into the study and as the Determine test gave negative result, no confirmatory test was done as suggested by the national algorithm. 

However, the statement on the HIV diagnosis has been rephrased to read “The two categories of participants, PLWH and HIV-negative individuals aged 1-72 years old were enrolled in the study and the HIV status of the HIV negative participants was determined using the Determine HIV Test strip.” to avoid misunderstanding. (See lines 134-135).

Also, I suggest that the inability to carry out further confirmation assays be included in a section with the subtitle "Study limitations". Other challenges/limitations encountered in the course of the study can also be included in this section.

We the authors believe that the “inability to carry out further HIV confirmatory test” is not a limitation, hence, should not be included in the “Study limitations” section. Based on the National algorithm, confirmatory test is definitely not a requirement for HIV negative participants tested by Determine 1. As for the 4 participants who were positive by Determine 1, their status was confirmed in the HIV Treatment and Care center following the standard operating procedure before they were included in the study. Hence the inability to carry out further confirmation is not a limitation as this was carried out.

---

## [Decision Letter · Decision Letter 6]

5 Jan 2021

PONE-D-20-20708R6

Malaria and intestinal parasite co-infection and its association with anaemia among people living with HIV in Buea, Southwest Cameroon: A community-based retrospective cohort study

PLOS ONE

Dear Dr. Sandie,

After careful consideration, we feel that your manuscript will likely be suitable for publication if the authors revise it to address additional points raised by the reviewer.  According to reviewer, there are some specific areas where further improvements would be of substantial benefit to the readers.   For your guidance, a copy of the reviewers' comments was included below. 

We look forward to receiving your revised manuscript.

Kind regards,

Luzia Helena Carvalho, Ph.D.

Academic Editor

PLOS ONE

Reviewers' comments:

Reviewer's Responses to Questions

**Comments to the Author**

1. If the authors have adequately addressed your comments raised in a previous round of review and you feel that this manuscript is now acceptable for publication, you may indicate that here to bypass the “Comments to the Author” section, enter your conflict of interest statement in the “Confidential to Editor” section, and submit your "Accept" recommendation.

Reviewer #3: (No Response)

2. Is the manuscript technically sound, and do the data support the conclusions?

Reviewer #3: Yes

3. Has the statistical analysis been performed appropriately and rigorously? 

Reviewer #3: Yes

4. Have the authors made all data underlying the findings in their manuscript fully available?

Reviewer #3: (No Response)

5. Is the manuscript presented in an intelligible fashion and written in standard English?

Reviewer #3: Yes

6. Review Comments to the Author

Reviewer #3: Pg 6, 134 - Pg 7, 144: From the authors' response to my review comments, I understood that the PLWH enrolled in this study were already confirmed cases undergoing treatment. Such patients undergo testing at intervals (determined by the physicians) to ascertain if their proviral load is still detectable. According to the authors, the confirmation tests were carried out by the HAART centres before the authors enrolled the participants. If that was the case, I suggest that authors include a statement clarifying that all confirmation tests were already performed by the treatment centres on the study participants before they were enrolled in the study, and that the Determine test was carried out by the authors perhaps for quality control purposes. I recommend that the statement be included in Pg 7, line 141, immediately after "PLWH  identified  through  the  outreach  programmes  not registered  in  the  HIV  treatment  centre  were  referred  to  the  treatment  centre  for  confirmation  and follow-up."

7. PLOS authors have the option to publish the peer review history of their article (what does this mean?). If published, this will include your full peer review and any attached files.

Reviewer #3: **Yes: **Nneoma Confidence JeanStephanie ANYANWU

---

## [Author Response · Author response to Decision Letter 6]

5 Jan 2021

Concerns or comments of the Reviewer:

Pg 6, 134 - Pg 7, 144: From the authors' response to my review comments, I understood that the PLWH enrolled in this study were already confirmed cases undergoing treatment. Such patients undergo testing at intervals (determined by the physicians) to ascertain if their proviral load is still detectable. According to the authors, the confirmation tests were carried out by the HAART centres before the authors enrolled the participants. If that was the case, I suggest that authors include a statement clarifying that all confirmation tests were already performed by the treatment centres on the study participants before they were enrolled in the study, and that the Determine test was carried out by the authors perhaps for quality control purposes. I recommend that the statement be included in Pg 7, line 141, immediately after "PLWH identified through the outreach programmes not registered in the HIV treatment centre were referred to the treatment centre for confirmation and follow-up."

For better understanding, a statement on HIV confirmatory test carried out at the Treatment centre was included in Pg 7, line 139 – 141 as “PLWH identified through the outreach programmes in the community were referred to the HIV care and treatment centre for confirmation of their HIV positive status and follow up before being enrolled among the PLWH category”.

---

## [Decision Letter · Decision Letter 7]

7 Jan 2021

Malaria and intestinal parasite co-infection and its association with anaemia among people living with HIV in Buea, Southwest Cameroon: A community-based retrospective cohort study

PONE-D-20-20708R7

Dear Dr. Sandie,

We’re pleased to inform you that your manuscript has been judged scientifically suitable for publication and will be formally accepted for publication once it meets all outstanding technical requirements.

Kind regards,

Luzia Helena Carvalho, Ph.D.

Academic Editor

PLOS ONE

Additional Editor Comments (optional):

Reviewers' comments:

Reviewer's Responses to Questions

**Comments to the Author**

1. If the authors have adequately addressed your comments raised in a previous round of review and you feel that this manuscript is now acceptable for publication, you may indicate that here to bypass the “Comments to the Author” section, enter your conflict of interest statement in the “Confidential to Editor” section, and submit your "Accept" recommendation.

Reviewer #3: All comments have been addressed

2. Is the manuscript technically sound, and do the data support the conclusions?

Reviewer #3: Yes

3. Has the statistical analysis been performed appropriately and rigorously? 

Reviewer #3: Yes

4. Have the authors made all data underlying the findings in their manuscript fully available?

Reviewer #3: Yes

5. Is the manuscript presented in an intelligible fashion and written in standard English?

Reviewer #3: Yes

6. Review Comments to the Author

Reviewer #3: (No Response)

7. PLOS authors have the option to publish the peer review history of their article (what does this mean?). If published, this will include your full peer review and any attached files.

Reviewer #3: **Yes: **Nneoma Confidence JeanStephanie ANYANWU

---

## [Editor Report · Acceptance letter]

11 Jan 2021

PONE-D-20-20708R7 

Malaria and intestinal parasite co-infection and its association with anaemia among people living with HIV in Buea, Southwest Cameroon: A community-based retrospective cohort study 

Dear Dr. Mekachie Sandie:

I'm pleased to inform you that your manuscript has been deemed suitable for publication in PLOS ONE. Congratulations! Your manuscript is now with our production department. 

Kind regards, 

on behalf of

Dr. Luzia Helena Carvalho 

Academic Editor

PLOS ONE